# Armor breakup and reformation in a degradational laboratory experiment

Clara Orrú[1], Astrid Blom[1], and Wim S. J. Uijttewaal[1]

[1]Department of Hydraulic Engineering, Faculty of Civil Engineering and Geosciences, Delft University of Technology, P.O. Box 5048, 2600 GA, Delft, The Netherlands.

*Correspondence to:* C. Orrú (C.Orru@tudelft.nl)

**Abstract.** Armor breakup and reformation was studied in a laboratory experiment using a trimodal mixture composed of a 1 mm sand fraction and two gravel fractions (6 and 10 mm). The initial bed was characterized by a stepwise downstream fining pattern (trimodal reach) and a downstream sand reach, and the experiment was conducted under conditions without sediment supply. In the initial stage of the experiment an armor formed over the trimodal reach. The formation of the armor under partial transport conditions led to an abrupt spatial transition in the bed slope and in the mean grain size of the bed surface, as such showing similar results to a previous laboratory experiment conducted with a bimodal mixture. The focus of the current analysis is to study the mechanisms of armor breakup. After an increase in flow rate the armor broke up and a new coarser armor quickly formed. The breakup initially induced a bed surface fining due to the exposure of the finer substrate, which was accompanied by a sudden increase of the sediment transport rate, followed by the formation of an armor that was coarser than the initial one. The reformation of the armor was enabled by the supply of coarse material from the upstream degrading reach and the presence of gravel in the original substrate sediment. Here armor breakup and reformation enabled slope adjustment such that the new steady state was closer to normal flow conditions.

## 1 Introduction

The formation of an armor has two different origins (e.g., Parker and Klingeman, 1982; Jain, 1990). A static armor is created when there is a lack of sediment supply and limited shear stress values enable entraining only the finer grains present at the bed surface. A mobile armor forms when a sediment mixture governed by a range in grain sizes is supplied from upstream. The over-representation of the coarse grains at the bed surface then serves to increase their transport rate. The coarsening of the bed surface of a static armor is mainly caused by the winnowing or washing out of fines from the bed (e.g., Parker and Klingeman, 1982; Dietrich et al., 1989). For a mobile armor the coarsening is mainly related to kinematic sorting or the infiltration of fines into the bed (e.g., Parker and Klingeman, 1982; Mao et al., 2011).

Armoring processes have been mainly investigated under controlled laboratory conditions. Many authors focused on the characteristics of the bed structure during armoring studying the bed arrangement and the stability of cluster particles (e.g., Church et al., 1998; Hassan and Church, 2000; Piedra et al., 2012; Heays et al., 2014). The stability of a gravel bed can be increased by the presence of cluster particles (e.g., Church et al., 1998; Hassan and Church, 2000; Piedra et al., 2012; Heays et al.,

2014). The armoring process is influenced by the nature of the sediment supply (e.g., Dietrich et al., 1989; Sklar et al., 2009). The experiments by Dietrich et al. (1989) showed that a decrease of the sediment supply rate corresponds to a more efficient armor development. Also the grain size distribution of the bed material and, in particular, the increase of sand content plays a role in the armoring process by reducing vertical sorting (Marion and Fraccarollo, 1997) or by reducing surface roughness

and facilitating the rearrangement of the bed particles (Curran and Waters, 2014). Another aspect is the influence of different flow rates on the development of the armor (Hassan et al., 2006; Guney et al., 2013). Hassan et al. (2006) indicated a general increase of the armoring degree with an increase of the flow rate. They also showed that temporally varying flows leads to stronger armoring than steady flows.

Still little is known on the behavior of an armor under peak flow (Vericat et al., 2006; Yager et al., 2015). Studies have demon-

strated that during peak flow armored surfaces can be stable and persist or broken and reform during the waning phase of the flood event. Andrews and Erman (1986) were the first to present a field case on armor persistence during peak flow. Using their surface-based transport model Wilcock and DeTemple (2005) predicted the persistence of an armor of a certain grain size distribution using transport rates measured in the field. Clayton and Pitlick (2008) described a gravel bed reach of the Colorado River with a persistent armor. The sediment supplied from upstream provided the sediment for the replacement of the entrained

particles. Grains from a persistent armor can exchange grains with the transported load (Wilcock and DeTemple, 2005). Despite these examples of armor persistence, there are also examples in which the armor does not persist during peak flow. Vericat et al. (2006) described armor breakup and reformation in a large river regulated by a dam. Their field measurements illustrated that the armor did not persist under peak flow and reformed at smaller flow values.

Various causes of armor breakup have been distinguished. The laboratory experiments by Klaassen (1988) provided detailed

measurements of armor breakup after a flood wave under bedform-dominated conditions. Here armor breakup appeared to result from the turbulence created in the trough zones of the migrating bedforms. A finer armor reformed in the waning phase of the flood wave. Wang and Liu (2009) conducted a laboratory experiment studying armor breakup under a shortage of sediment supply. The armor was created under base flow and its stability was tested under a stepwise increase of the discharge. Armor breakup was due to an increased mobility of the coarse particles and led to a sudden increase of the bed load transport.

Other causes of armor breakup are the sediment supply from upstream and the presence of bedload sheets. The supply of finer material can lead to an increased mobility of the coarse sediment and can therefore mobilize the armor (Sklar et al., 2009; Venditti et al., 2010a, b; Spiller et al., 2012). The presence of bedload sheets can reduce armor stability by reducing the bed roughness and increasing the flow velocity (Iseya and Ikeda, 1987; Kuhnle and Southard, 1988; Recking et al., 2009; Bacchi et al., 2014). The conditions and parameters that determine the persistence or breakup of an armor are still unclear. One of

the problem is the fact that it is difficult to measure temporal changes of the bed surface texture during peak flow in the field (Wilcock and DeTemple, 2005).

The objective of this paper is to study the mechanisms of armor breakup and their consequences by providing detailed measurements of spatial and temporal changes of the bed surface texture under controlled laboratory conditions. The changes of the bed surface texture were measured during flow using the technique developed by Orrú et al. (2014, 2016). We examine

the stability of an armor under peak flow under a limited sediment supply. The experimental set-up was characterized by an

initial streamwise fining pattern, following a previous laboratory study on armor formation in which a reach characterized by an initially gradual fining pattern under a limited sediment supply rate and partial transport conditions, developed into a more abrupt spatial transition in grain size and slope (Orrú et al., 2016).

The results presented by Orrú et al. (2016) suggest a similarity with a gravel-sand transition (e.g., Yatsu, 1955; Shaw and Kellerhals, 1982; Parker, 1991a, b; Frings, 2011; Venditti et al., 2015), yet the mechanisms prevailing in the experiment were likely not comparable to the ones governing natural gravel-sand transitions. Relevant mechanisms in the development of a gravel-sand transition are the progradation of a gravel wedge (Paola et al., 1992; Seal et al., 1997), basin subsidence (Paola, 1988; Parker and Cui, 1998), base level change (Pickup and Warner, 1984; Sambrook Smith and Ferguson, 1995), and suspended load (Venditti and Church, 2014; Venditti et al., 2015), which do not play a role in the Orrú et al. (2016) experiment as well as in the current laboratory experiment. Nevertheless, the current experiments may provide insight in the co-existence of an armor and a sand patch during peak flow.

## 2    Experimental set-up

### 2.1    Experimental settings

The experiment was carried out at the Water Laboratory of the Faculty of Civil Engineering and Geosciences of Delft University of Technology. The experiment was conducted in a tilting flume that was 14 m long, 0.40 m wide, and 0.45 m high. The upstream water supply was controlled by a water pump and the downstream water level was set by a tailgate located at the downstream end of the flume. No sediment was supplied from upstream. At the downstream end of the flume the transported sediment was collected in a sediment trap.

We used a trimodal sediment mixture that was composed of a sand fraction ($D_{50,1} =1$ mm) and two gravel fractions, a medium fraction ($D_{50,2} =6$ mm) and a coarse fraction ($D_{50,3} =10$ mm). The sediment fractions were painted in different colors to enable measurements of the grain size distribution of the bed surface using the image analysis technique of Orrú et al. (2016). The fine fraction was left with its natural color, the medium fraction was painted yellow green, and the coarse fraction was painted medium turquoise.

As mentioned above the experimental set-up was similar to the one by Orrú et al. (2016). Here we use a trimodal mixture rather than a bimodal one. The initial bed was installed with an imposed stepwise fining pattern. The upstream reach was composed of the trimodal mixture (the trimodal reach) and the downstream reach was composed of sand (the sand reach). The trimodal reach was characterized by 10 compartments characterized by a length of 0.40 m (Fig. 1), and the length of the most upstream compartment was 0.88 m. Each compartment was characterized by a different initial volumetric fraction content of the 3 fractions (Fig. 1). The sand content increased in streamwise direction in steps of 10 percent for each compartment. The bed slope was set equal to 0.0022. We refer to Orrú et al. (2016) for details regarding the method to install the initial bed.

An initial experiment (T1) was conducted to create an armor under base flow conditions. The flow conditions were increased in Experiment T2 to assess the stability of the armor under peak flow (Fig. 2). The flow regime was subcritical in both experiments. During the initial experiment the water discharge was equal to 0.0465 $m^3s^{-1}$. The downstream water surface elevation was

adjusted during the first few flow hours and maintained constant for the remainder of the experiment. The total duration of Experiment T1 was 16 hours, and the time interval of experiment T1 is from -16 h to 0 h. At the beginning of Experiment T2 the water discharge was set equal to $0.0547 \text{ m}^3\text{s}^{-1}$ and the downstream water surface elevation was decreased through lowering the tailgate. For the remainder of the experiment water discharge and water surface elevation were maintained constant (Fig. 2).

Experiment T2 lasted 4 hours, and the time interval of Experiment T2 is from 0 h to 4 h.

## 2.2 Measurements

The water discharge was constantly measured at the input water pipe using a flow meter that measured the travel time of an acoustic signal. Two laser instruments mounted on a carriage were used to measure longitudinal profiles of the bed and water surface elevations at the center of the flume approximately every 20 minutes. The laser instrument used to measure bed

elevation was placed in a watertight eye-shaped box that was slightly submerged to avoid reflections of the signal on the water surface. At the downstream end of the flume (at x = 10.62 m) the water surface elevation was continuously measured using a pitot tube. A linear position sensor was connected to the pitot tube by a hose and positioned besides the flume. The transported sediment was caught in a sediment trap at the downstream end of the flume. The sediment was pumped to a small tank which was placed on a scale for measuring the submerged sediment mass.

The grain size distribution of the bed surface was measured during flow over the entire observation section ($\approx$10 m). The measurements were taken using the image analysis technique developed by Orrú et al. (2014, 2016), which is based on color segmentation, i.e. the division of the pixels of an image into color groups (Fig. 3). To this end each grain size fraction was painted in a different color. The equipment used to take the images of the bed surface was composed of a carriage that enabled moving the equipment along the flume and a floating device (Fig. 4). A camera was connected to the carriage. The floating

device, which was slightly submerged, was connected to the carriage using PVC pipes. The lower pipes had a smaller diameter than the upper ones to allow for vertical motion of the floating device and automatic adjustment of the level of the floating device to spatial changes in the water surface elevation. The design and material of the floating part of the measurement equipment were here optimized compared to the one presented by Orrú et al. (2016) to reduce its submersion. For this purpose the bottom of the upstream V-shaped part of the floating device was designed with a small inclination to obtain a lift force from

the flow. The floating device was made out of thin transparent Plexiglas®. Six small LED lights were mounted on the frame of the floating device to illuminate the bed surface. The images of the bed surface were processed using a Matlab algorithm that provided the areal fraction content of a surface covered by a certain color (i.e., grain size). The areal fraction contents resulting from image analysis were converted into volumetric fractions applying the conversion model of Parker (1991a, b).

## 3 Formation of the initial armor (Experiment T1)

In this section we briefly describe Experiment T1 conducted to create the armor under the imposed supply limited conditions. Over the trimodal reach the initially high rate of sand entrainment combined with the slightly mobile gravel fractions quickly resulted in a coarse bed surface (Fig. 5). A limited amount of sand was available at the bed surface. In the remaining part of

Experiment T1 the coarser fractions were less or no longer mobile (partial transport conditions) due to the formation of an armored bed structure that enhanced particle stability. The prevailing mechanisms were winnowing and the kinematic sieving of sand. Bed surface coarsening was observed between x = 1 m and x = 4.5 m (Fig. 5). Some randomly arranged irregularities over the armor created gaps between the gravel particles where the finer substrate was exposed.

The armoring occurring over the trimodal reach limited the sediment supplied to the sand reach, which resulted in a strong bed degradation over the sand reach (Fig. 6), as was observed by Orrú et al. (2016). This spatial difference in degradation resulted in a sudden decrease of the bed elevation between the trimodal and the sand reach. Adjusting to limited sediment supply conditions the bed approached a final state that was characterized by zero sediment transport. For the sand reach, the state of zero sediment transport was governed by a much smaller flow velocity (and so larger flow depth) than for the

upstream trimodal reach, which resulted in the observed step in bed elevation. This sudden decrease in bed elevation resulted in a streamwise increase in the water surface elevation, which is a Bernoulli effect (e.g., Douglas et al., 2005). The trimodal reach of Experiment T1 was characterized by the presence of an M1 backwater curve due to the different bed slopes and so flow depths between the two reaches. The final stage of the bed of Experiment T1 was characterized by an abrupt transition in slope and bed surface texture between the two reaches (Fig. 5).

The upstream section of the trimodal reach was governed by an imbricated structure (Fig. 7). This structure formed in the initial part of the experiment when the gravel fractions were still slightly mobile. The particles quickly found a stable position that enhanced the armor stability. The armor was considered fully developed after 16 flow hours (i.e., at time 0 h) when no relevant changes of the bed surface texture were observed and the sediment transport rate reached approximately zero.

## 4   Breakup and reformation of the armor layer (Experiment T2)

### 4.1   Bed surface texture

At the beginning of the armor breakup experiment, Experiment T2, the flow velocity was increased by increasing the water discharge by 18 % and lowering the downstream water level (Fig. 2). Armor breakup and reformation covered a short period. After the increase of the flow velocity the armor started to break up in several sections of the trimodal reach. In the upstream part of the trimodal reach the substrate was coarser due to a limited amount of sand and the bed was highly imbricated (Fig. 7).

This imbrication enhanced armor stability and consequently some sections of the armor did not break up. Figure 8 shows a section of the trimodal reach where some part of the armor was broken. Initially, the dislodgement of a few gravel particles enabled the entrainment of the finer subsurface material over a small section of the bed (Fig. 8a). Subsequently, the sand entrainment appeared to enhance the gravel mobility, which may have extended the breakup exposing the subsurface over a wider section (Fig. 8b). Blurriness in the images shown in Figures 8a, b indicates the entrainment of sand. The measurements

of the bed surface texture show a fining of the bed between x = 1 m and x = 4.5 m (Fig. 9- 10). The measurement taken after 7 min (point 3 in Fig. 9) corresponds to the bed state of Figure 8c. As we observed that the bed surface at the moment of the breakup was finer(Fig. 8b), we hypothesize that the bed surface at the moment of the breakup was finer (point 2 in Fig. 9) than the moment of Figure 8c (point 3 in Fig. 9).

The breakup and bed surface fining were quickly followed by the formation of a mobile armor that was coarser than the initial one (point 4 of Fig. 9). The coarse sediment supplied from upstream enabled the formation of a new armor and the presence of gravel particles in the substrate aided this armor reformation. After armor reformation a limited amount of sand was present at the bed surface (Fig. 9). The fact that the reformed armor was slightly coarser than the initial one resulted in a slight downstream

coarsening in the gravel reach.

## 4.2  Bed elevation

The total amount of degradation due to armor breakup depended on the texture of the substrate material. In the upstream part of the trimodal reach the substrate was coarser which limited bed degradation. We observed a lateral variation in degradation characterized by a stronger degradation at one side of the flume that is not evident in our measurements since bed profiles were

measured only at the center of the flume. The breakup led to a fast degradation which was arrested by the reformation of the mobile armor (Fig. 6). The degradation was not uniform in streamwise direction. Over the reach that suffered from the breakup the slope decreased to adjust to a situation with a shortage of sediment supply. The redistribution of the sediment led initially to aggradation downstream of the breakup area and subsequently to the progradation of the front between the trimodal reach and the sand reach (Fig. 6). The progradation had ceased at the end of the experiment.

The decrease of the slope made the bed locally approach normal flow conditions, which indicates a state in which the slope of the water surface is equal to the slope of the channel bed. If boundary conditions of a reach (i.e., upstream water discharge and sediment supply rate, as well as the downstream water surface elevation) are constant for a sufficiently long time, it will reach normal flow conditions (provided that particle abrasion, tributaries, and subsidence or uplift do not play a role). Yet, under conditions of partial transport, in which the coarse fractions of the sediment are immobile, the associated armor can prevent

this adaptation of the bed and its approach to normal flow conditions (see Experiment T1 and Orrú et al. (2016)). Here armor breakup enabled adjustment of the bed slope such that the bed slope became closer to the water surface slope and the final bed configuration was closer to normal flow.

During Experiment T1 and T2 the hydraulic conditions not only varied spatially but also temporally (Fig. 11). The adjustment to the limited sediment supply led to a coarsening over the trimodal reach and a larger flow depth over the sand reach (Fig. 11a).

Due to the temporal increase of the water discharge and a lowering of the downstream water surface elevation, flow velocities were increased in Experiment T2 relative to Experiment T1. The highest flow velocities were present at the moment of armor breakup (Fig. 11b). Both experiment were characterized by a subcritical flow regime (Fig. 11c). Shields stress values for each sediment fraction were determined accounting for the sidewall correction of Vanoni and Brooks (1957). Highest Shields stress values over the trimodal reach were observed when the armor broke up (Fig. 11d, e, f). Despite the high Shields stress values

observed over the entire trimodal reach, the breakup was local and not uniform over the reach. Reformation of the mobile armor was associated with a decrease of the Shields stress (Fig. 11d, e, f).

Figure 12 shows the local sediment transport rate computed from the migration of the front between the trimodal reach and the sand reach (Fig. 12a), as well as the sediment transport rate measured at the downstream end of the flume (Fig. 12b). We determined the local sediment transport rate at the position of the front, $q_{front}$, from the streamwise migration speed of the

front as proposed by Bagnold (1941) using the simple-wave relation:

$$c = \frac{q_{front}}{c_b \Delta} \tag{1}$$

where $c$ denotes the migration speed of the prograding front in streamwise direction (here determined at the crest), $c_b = 1 - p$, $p$ being the bed porosity (we assume $p = 0.4$) and $\Delta$ denotes the height of the prograding front.

The sediment load at the front was composed of the 3 grain size fractions, whereas the sediment load at the downstream end of the flume was composed of mainly the sand fraction. The transport rates at the front and downstream end of the flume are of the same order of magnitude (Fig. 12). At both locations the sediment transport rate shows a (sudden) increase, which was followed by a gradual decrease. The peak in the sediment transport rate at the downstream end of the flume shows a time lag. Also the decrease in the sediment transport rate was slower than at the front. The sediment transport rate at the front increased

as a result of the armor breakup. This increase at the front coincided with the rapid entrainment of the substrate material and the decrease of the sediment transport rate corresponded to the reformation of the mobile armor. Yet, the sediment transport rate at the downstream end seemed to be less affected by the breakup and seemed to respond to primarily the increased flow rate.

## 5    Discussion

Our experiment indicates that armor breakup and reformation can be a fast process and that the resulting changes in bed elevation may be limited. Information on the time scales of armor breakup and reformation and the order of magnitude of resulting bed elevation changes may be relevant to flood risk and navigation studies over armored reaches, for instance in the upstream part of the IJssel branch of the Dutch Rhine. The dislodgement of armor particles may expose and release fine sediment from the substrate and such sudden supply of fine sediment may result in local aggradation in the downstream reach

that creates problems to navigation. The stability of an armor is also of interest to the design of granular filters aimed at protecting structures from scour and in the operation of dams, for instance in the design of flushing flows undertaken to release sediment from a reservoir.

It is still difficult to predict armor breakup. This is due to (a) the possible randomness associated with the position of the breakup, and (b) the fact that its mechanisms are not sufficiently clear. Here and in the studies by Vericat et al. (2006) and Wang

and Liu (2009) the increase of the flow discharge led to an increase in the sediment mobility, which caused the armor to break up. The results of our experiment showed an almost uniform increase of the Shields stress over the trimodal reach, however the breakup was local and its position seemed to be random. Let us consider causes additional to an increased sediment mobility that may have played role in the breakup process in our experiment. Sediment supplied from upstream may induce breakup by destabilizing the armor (Spiller et al., 2012). Such mobilization has been encountered when finer material is supplied to the

armor surface (Sklar et al., 2009; Venditti et al., 2010a, b; Spiller et al., 2012). By filling the gaps of the coarse surface the fine sediment reduces the bed friction, which increases the flow velocity. A similar process occurs when bed friction is reduced due to the transport of finer material in bedload sheets (Iseya and Ikeda, 1987; Kuhnle and Southard, 1988; Recking et al.,

2009; Bacchi et al., 2014). These potential causes can be ruled out in our experiment because the material supplied from the upstream slightly degrading section was mainly coarse. We expect that the destabilization of the armor may also be ascribed to the impact of transported particles onto the gravel particles that are at rest. Klaassen (1988) attributed the destabilization of the armor in his experiments to turbulence originated by migrating bedforms. In our plane bed experiment additional turbulence

may have been created by irregularities at the armor surface. Turbulence and the resulting pressure fluctuations due to these irregularities may have caused or increased entrainment of sand from the sandy substrate in the more downstream part of the trimodal reach, which may have enhanced gravel mobility and facilitated the lengthening of the breakup.

The moment of the breakup was characterized by an increase of the bed load transport rate, which was also observed in the laboratory experiments of Klaassen (1988) and Wang and Liu (2009). The increase in the sediment transport rate was rapid

and sudden and corresponded to the mobility of the armor particles and the entrainment of the finer substrate material. A temporal fining of the bed surface characterized the moment of the breakup and it was followed by a coarsening due to the reformation of an armor that was coarser than the initial one. Similar results were presented by Vericat et al. (2006), however in their case armor reformation occurred only under base flow. In their field case the increased degree of armoring was caused by partial transport conditions. In our experiment the armor reformed under continued peak flow conditions. The coarser upstream

section acted as a source of sediment for the finer downstream reach. This supply provided the replacement for the entrained material, likely resulting in a quick reformation of the armor. The gravel present in the original substrate may have aided armor reformation. Possible causes of the fact that the reformed armor was coarser than the initial one are: (a) the supply from upstream being mostly gravel, (b) the sand supplied from upstream not being trapped in the zone of the broken up armor, and (c) the higher flow rate not allowing for the sand to deposit and remain between the gravel particles.

Similar to the experiment of Orrú et al. (2016), Experiment T1 showed the development of a reach characterized by a gradual fining pattern and uniform slope into an abrupt transition in bed surface grain size and slope as encountered in natural gravel-sand transitions (e.g., Yatsu, 1955; Shaw and Kellerhals, 1982; Parker, 1991a, b; Frings, 2011; Venditti et al., 2015). Here Experiment T2 provides new insights in the co-existence of an armor and a sand reach during peak flow such as: the starvation and degradation of the sand reach, progradation of the gravel front, impact of a reduced slope on the progradation of the gravel

front and armor reformation.

## 6   Conclusions

A flume experiment was conducted to investigate the stability of an armor under conditions with a limited sediment supply. The armor formed over a bed initially characterized by a gradual fining pattern under base flow conditions. The armor broke up after an increase of the flow rate and rapidly reformed under continued peak flow conditions. Despite the fact that the Shields

stress almost uniformly increased over the trimodal reach, the breakup was local and not uniform over the reach. Besides the increased flow rate multiple factors may have contributed to the armor breakup such as the impact of coarse sediment supplied from upstream and turbulence created due to irregularities of the armored surface.

The breakup was characterized by a temporal fining of the bed surface due to local degradation and the exposure of finer sub-

strate sediment. Despite the limited sediment supply conditions after the armor breakup a new armor, which was coarser than the initial one, quickly formed. Coarse sediment supplied by the upstream degrading reach provided the sediment required for the armor to reform, which was aided by the gravel in the substrate sediment. Armor breakup coincided with a sudden and local increase of the sediment transport rate due to the entrainment of the finer substrate material. This was followed by a gradual

5 decrease, which corresponded to the armor reformation. The breakup led to a decrease of the Shields stress and the local bed slope.

Partial transport conditions can prevent the adjustment of the bed and the approach to normal flow conditions (i.e., the equilibrium state was characterized by a backwater). Here armor breakup enabled the adjustment of the bed slope such that the final bed configuration was closer to normal flow, i.e., the bed slope was closer to the water surface slope.

10 *Acknowledgements.* The authors especially thank the technicians of the Water Laboratory of Delft University of Technology for their assistance during the experiments.

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

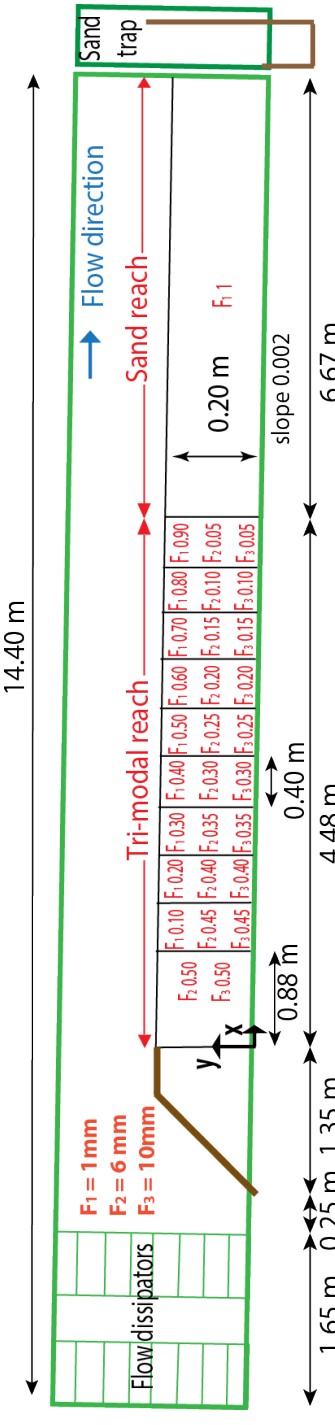

**Figure 1.** Flume set up and initial bed for Experiment T1. The red numbers indicate the volumetric gravel fraction content in each compartment.

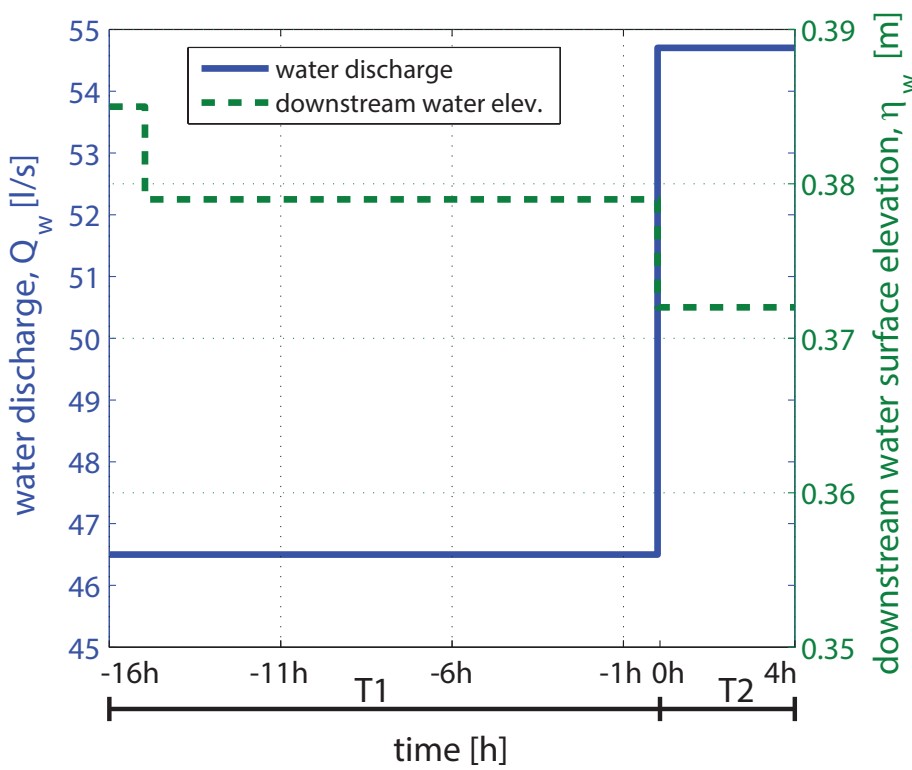

**Figure 2.** Water discharge, $Q_w$, and base level (i.e., water surface elevation at the downstream end), $\eta_w$, imposed in Experiments T1 and T2. The water surface elevation was measured at x = 10.62 m.

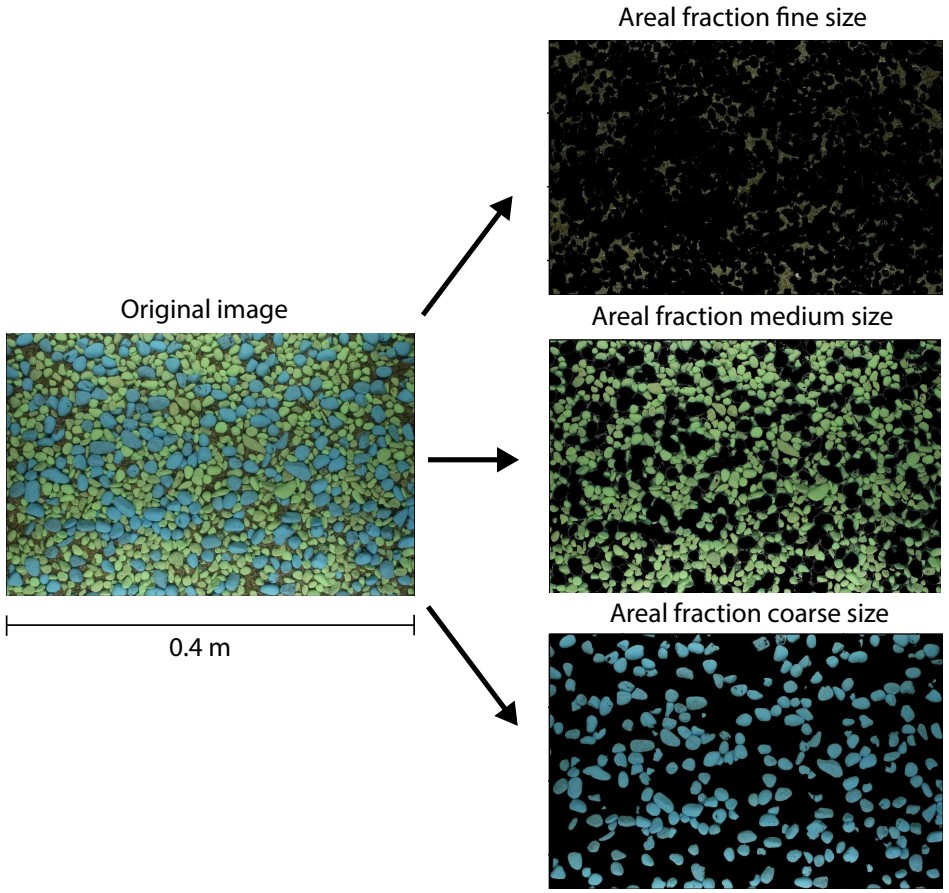

**Figure 3.** Example of color segmentation for an image of the bed surface applied to determine the grain size distribution of the bed surface.

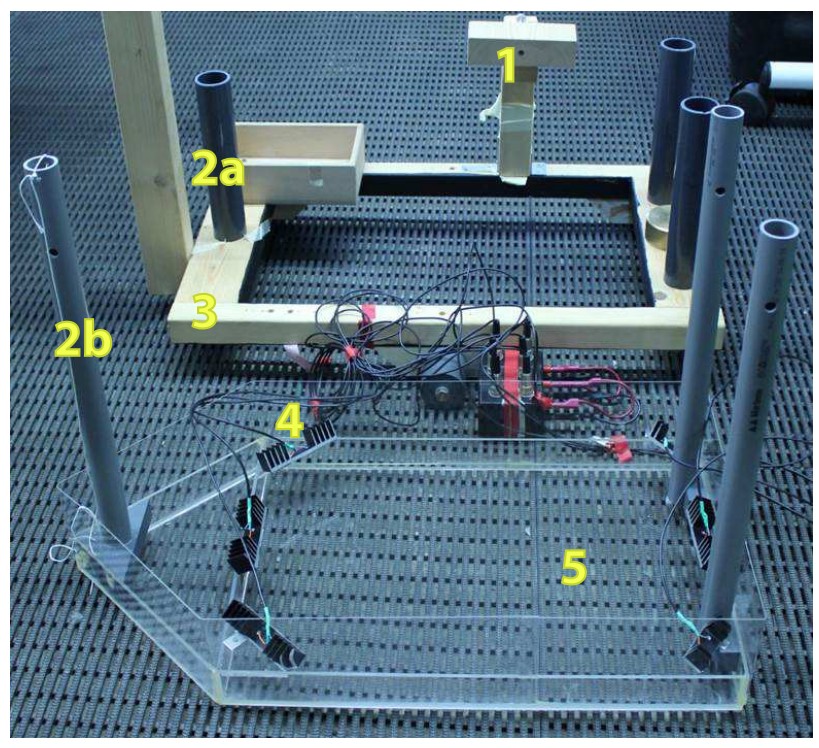

**Figure 4.** Floating part of the equipment for measuring the grain size distribution of the bed surface during the laboratory experiments. Here the bottom and the walls of the floating part are made of thin Plexiglas® plates and higher Plexiglas® sheets are attached along the walls to avoid water overflow. The numbers indicate: (1) position of the camera, (2a) upper pipe, (2b) lower pipe, (3) carriage, (4) LED light attached to the cooling plate, and (5) floating part.

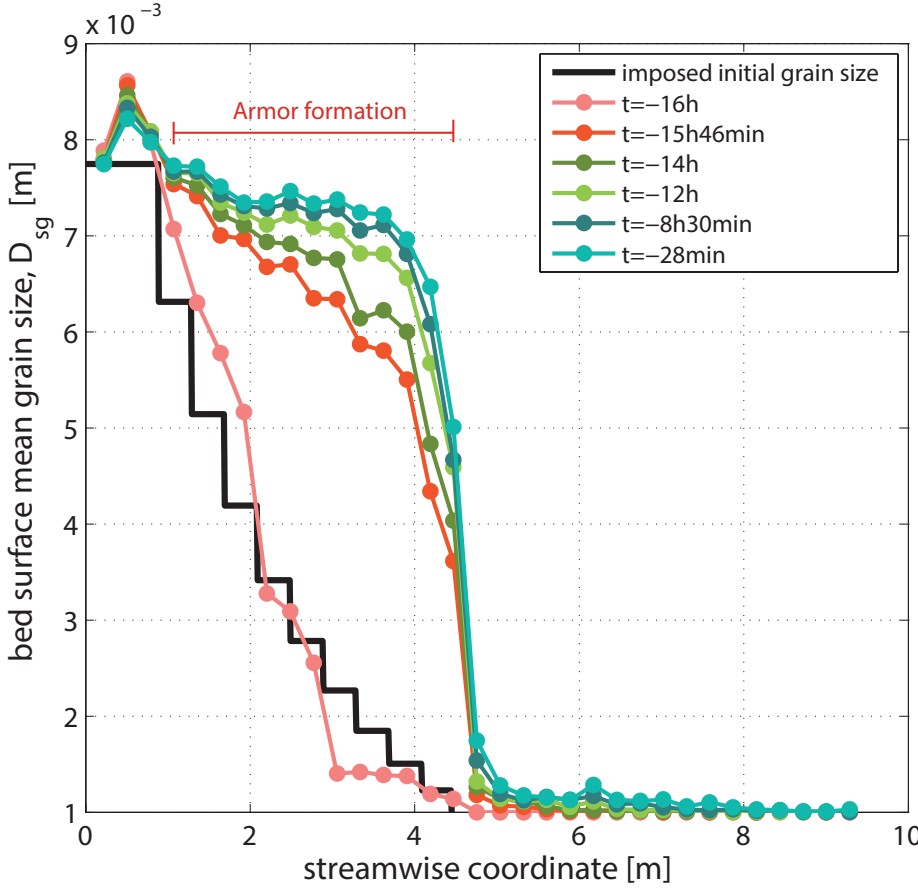

**Figure 5.** Imposed and measured geometric mean grain size of the bed surface sediment at various times for Experiment T1.

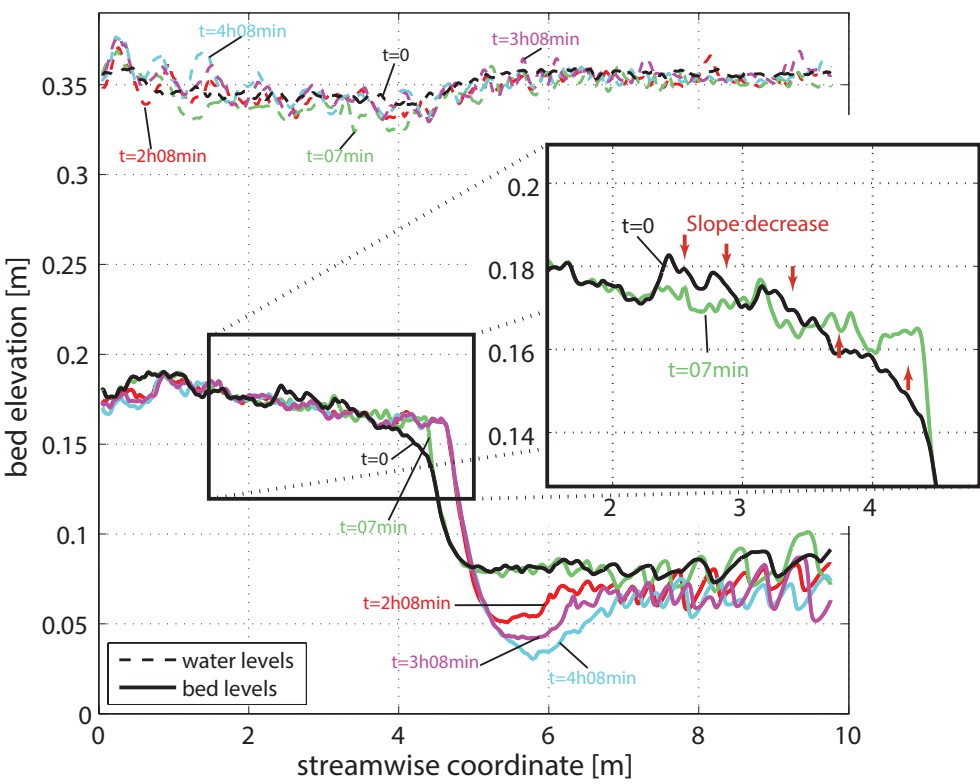

**Figure 6.** Measured water surface and bed elevation profiles of Experiment T2 at t=0h, 07min, 2h08min, 3h08min, and 4h08min. The zoomed window shows the degradation occurring during the armor breakup. Flow is from left to right. The profiles were smoothed using a Gaussian filter.

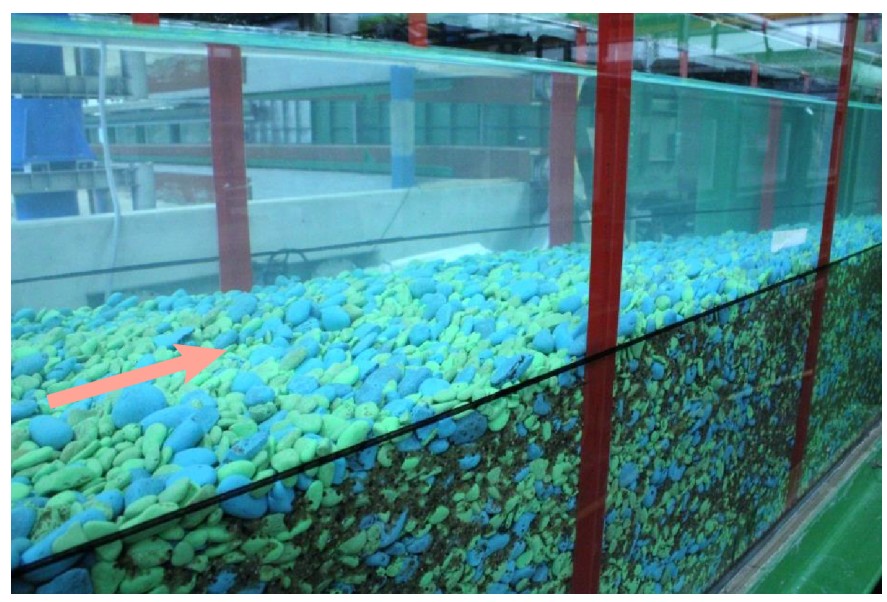

**Figure 7.** Imbrication of the bed surface sediment at the end of Experiment T1. Flow is from left to right.

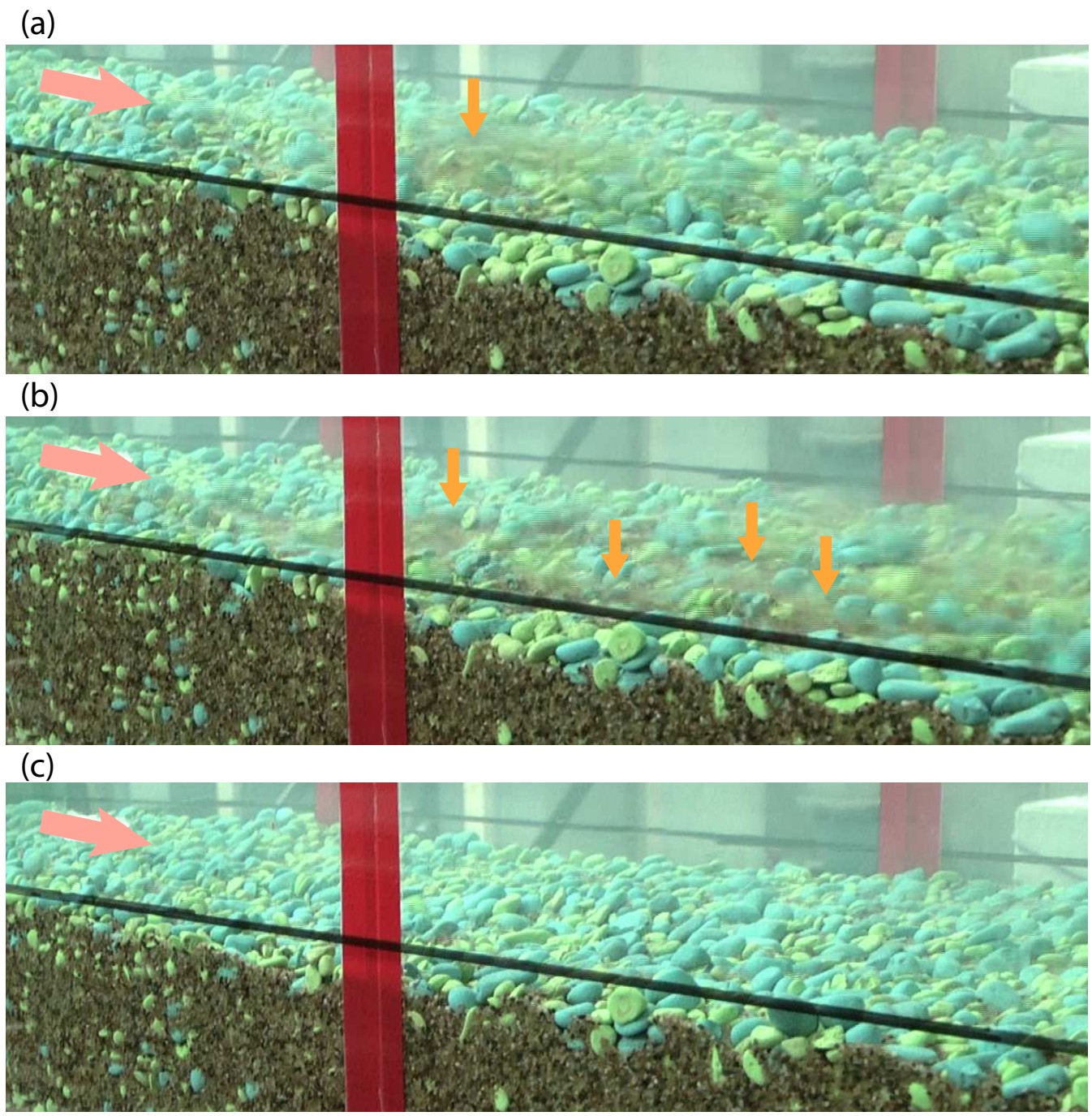

**Figure 8.** Armor breakup and reformation between Compartments 6 and 7: (a) initial local breakup, (b) widening of the breakup and exposure of the finer substrate, (c) reformation of the mobile armor. Flow is from left to right. Blurriness in images (a) and (b) indicates the entrainment of sand (orange arrows).

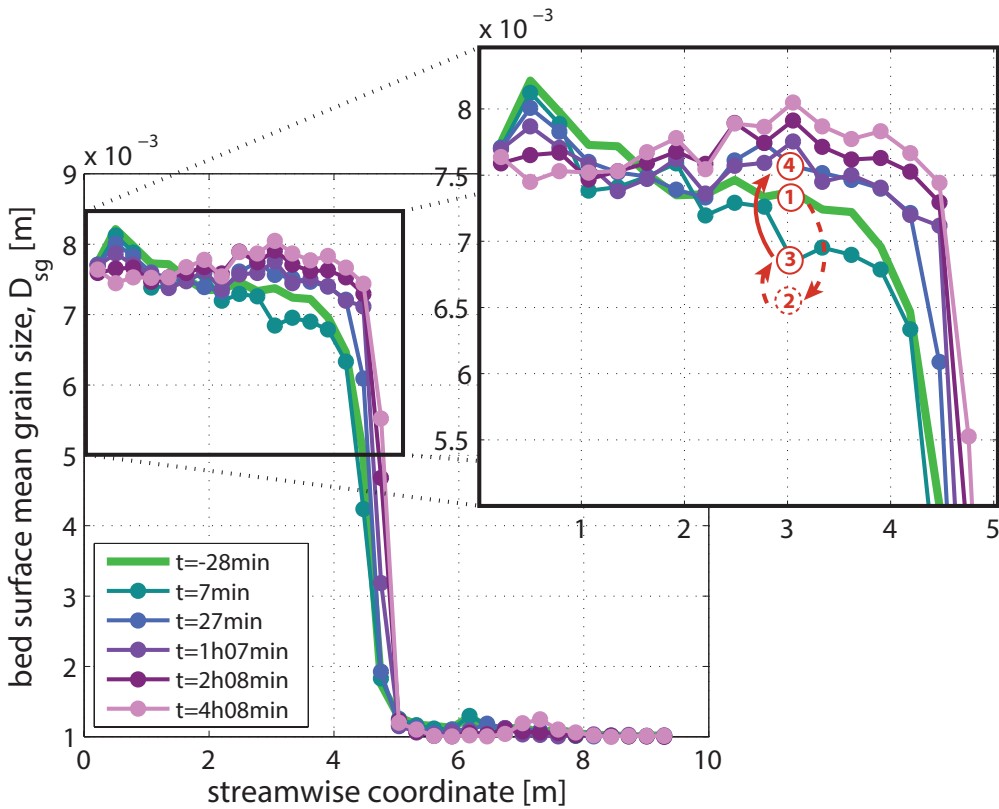

**Figure 9.** Measured geometric mean grain size of the bed surface sediment at various times for Experiment T2. Point 1 to 4 in the zoomed window show the temporal change of the bed surface. Point 2 indicates a hypothetical (i.e., not measured, see Fig. 8b) finer surface at the moment of the breakup; point 3 corresponds to the coarser bed surface shown in Fig. 8c right after the reformation of the armor.

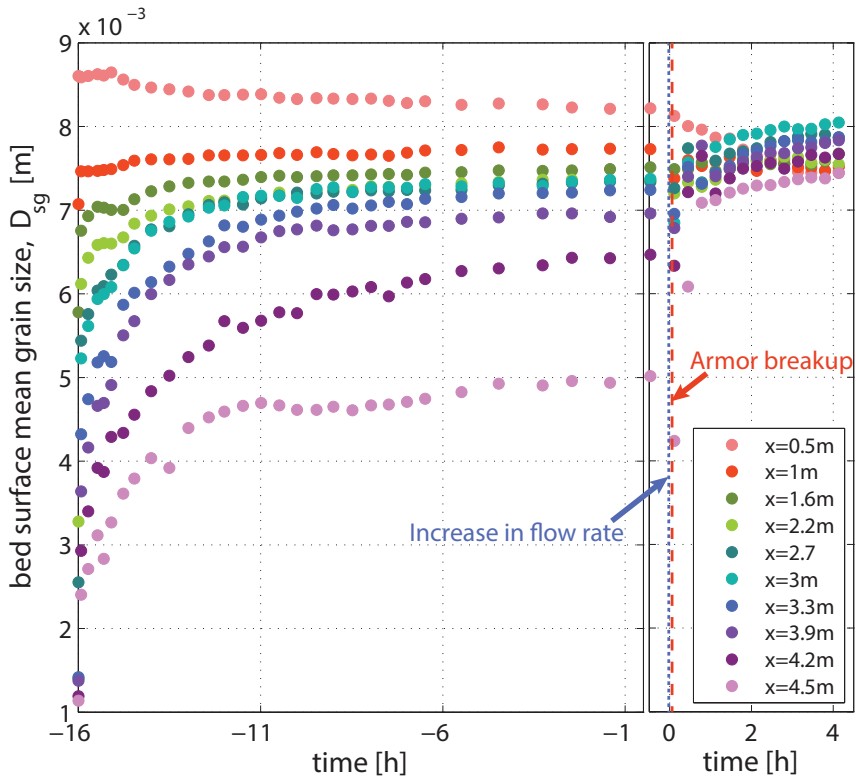

**Figure 10.** Measured change of the geometric mean grain size of the bed surface sediment at various locations in Experiments T1 and T2. The blue dotted line indicates the moment of the increase in flow rate (t=0 h) and the red dashed line indicates the moment of armor breakup (t=4 min).

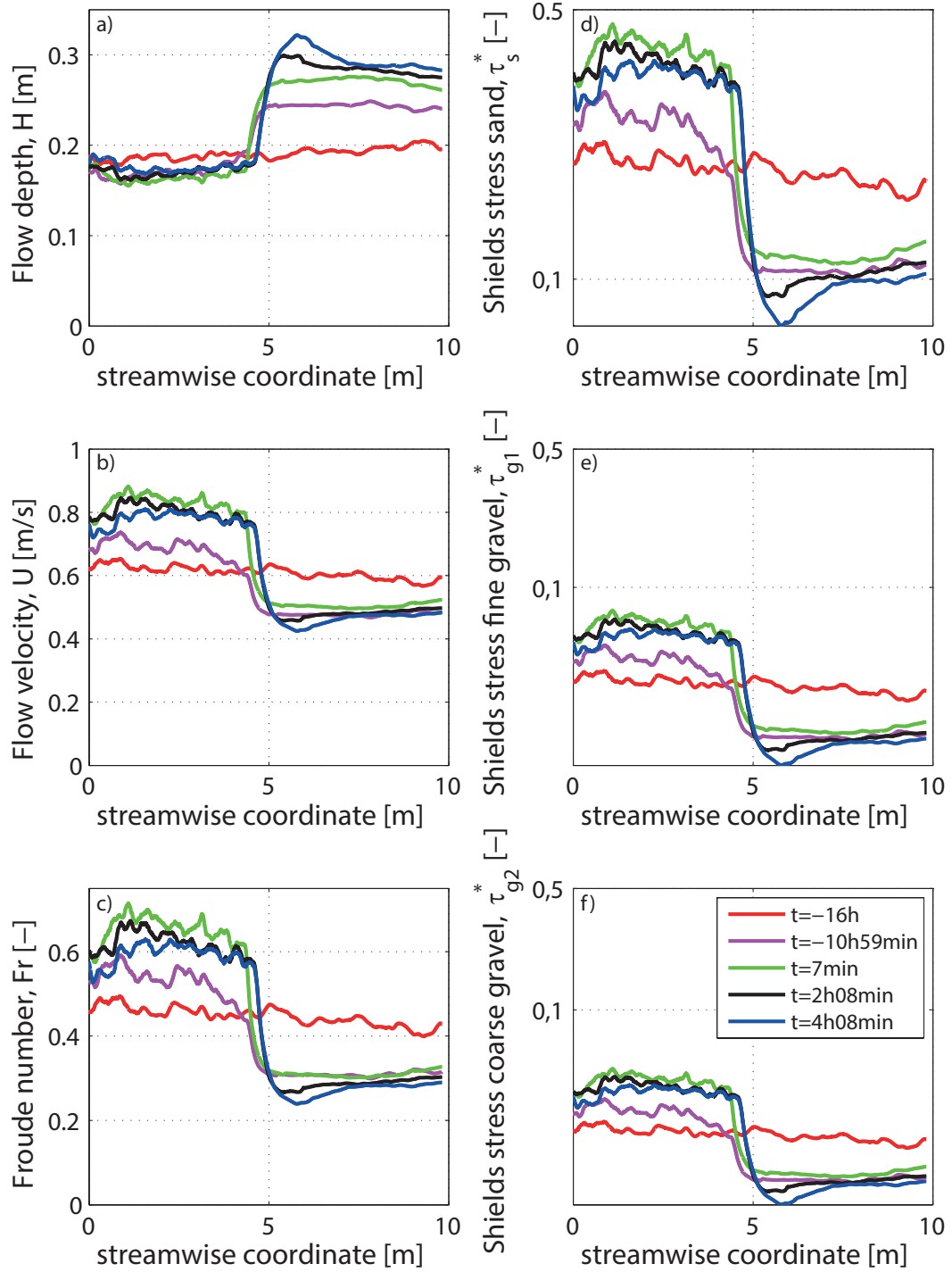

**Figure 11.** Hydraulic conditions of experiment T1 and T2: (a) flow depth, (b) flow velocity, (c) Froude number, (d) Shields stress for the sand fraction (logarithmic scale), (e) Shields stress for the fine gravel fraction (logarithmic scale), (f) Shields stress for the coarse gravel fraction (logarithmic scale). The Shields stress values were determined accounting for a sidewall correction due to Vanoni and Brooks (1957).

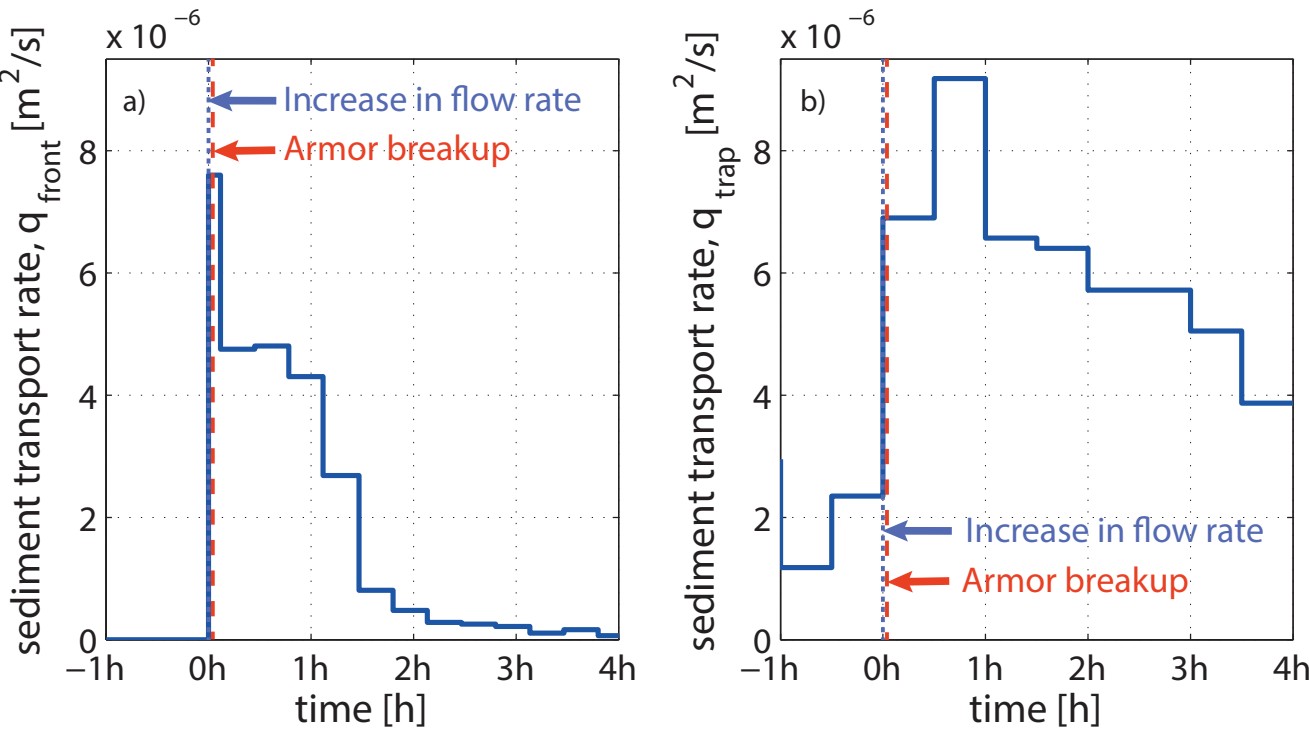

**Figure 12.** Measured sediment transport rate (a) computed from the migration of the front between the trimodal reach and the sand reach using Eq. (1), (b) at the downstream end of the flume. Horizontal line intervals indicate time-averaged values.