# Peer review of "Armor breakup and reformation in a degradational laboratory experiment"

_Earth Surface Dynamics, 2016_

## Referee Comment (RC1) · Anonymous Referee #1 · 7 Feb 2016

SUMMARY

This paper examines how the texture of a mixed sand-gravel channel bed responds to changes in flow discharge. It is qualitatively interesting to see the formation of static armor during high flow followed by breakup of this armor and reformation of a coarser mobile armor during subsequent high flow. Furthermore, application of new techniques for repeated longitudinal profiles of grain size and bed elevation offers new quantitative insight into armoring processes.

While some of the observations are interesting, I think they are insufficient to merit

publication. Though it was very long (20 h), only one experiment was run. Therefore, it is impossible to determine the reproducibility of these results or the dependence of the armor formation and breakup processes on the specific initial bed configuration, pattern of flow changes, and trimodal bed texture. Additional experiments to vary at least one of the experimental variables (e.g., magnitude of low and high flow) would provide much more insight into the controlling factors of bed armoring.

Furthermore, the manuscript is poorly organized and needs to be fundamentally restructured. The abstract and conclusion are nearly identical and do not establish the motivation and implications for this work. The introduction is very disorganized and does not follow a logical progression in presenting information about past work. For example, the opening sentence of the paragraph at line 18 on page 1 seems unrelated to the remaining content in the paragraph. Some information in the methods (section 2) really belongs in the results (section 3); e.g., section 2.3. In the results, there are several assertions made without proper explanation. For example, on page 4, line 14-15, an "imbricated structure" is mentioned, but there is never any discussion of when the imbrication developed or what morphological features suggest this interpretation. Related to this, there are several interpretations in the results (section 3), which really belong in the discussion (section 4); e.g., section 3.2, lines 9-10. Finally, all figures should be mentioned in the manuscript (currently Fig. 2 is missing), and the figure numbering should correspond to the order of mention (currently Fig. 10 is mentioned before Fig. 6).

Given these concerns, I therefore recommend rejection for this manuscript. I encourage the authors to run additional experiments, then more clearly present their results for future publication of this work. Further minor comments are listed below.

MINOR COMMENTS

1. In the abstract, the phrasing, "trimodal mixture composed of sand and gravel," implies that there are three components, but only two are mentioned here. It would be

better just to list the three sizes (1, 6, and 10 mm) here.

2. Page 3, line 13-17. Why do you impose a stepwise fining pattern in the initial bed? How does this reveal the dynamics of static and mobile armor formation better than a uniformly graded bed? Please explain in the manuscript. Also, the description of patch lengths is confusing. You should refer here to Fig. 2, which is never mentioned in the text.

3. Section 2.2 on grain size techniques is insufficient to understand these techniques. Unless you have a strong reason to delve into the details, I suggest summarizing this to one or two lines then referring to Orru et al (submitted 2015) for further information. I also suggest removing Figures 3 and 4 for this reason.

4. In describing the armor breakup and reformation, it is very difficult to see these bed changes from image to image in Figure 7. Could you apply some kind of image differencing technique to make the changes more apparent? The quantitative information in Figs. 8 and 9 is much more useful. Finally, unless you can justify them with more quantitative information, I would suggest removing the assertions on page 4, line 27-29, about textural changes between grain size analyses.

5. The methods used for characterizing the bedload transport rate from the front propagation are confusing (page 5 and Fig. 11). Where is the front being measured, in terms of the streamwise coordinate? Is there really no observed transport here prior to the step increase in discharge (as indicated in Fig. 11), or are you just assuming this?

6. At page 5, line 26, it is not clear whether the mentioned bed load transport rate is referring to the propagating front bed load transport or some other measurement. It is curious that a bed load trap is mentioned on page 2, lines 32-33, but then never again mentioned in the paper.

7. Finally, I am a bit concerned that no mention of the shear stress or Shields parameter is made here, despite the fact that this is usually considered an important variable in

studies of the evolution of mixed sediment surfaces and armoring (e.g., Wilcock and Crowe, 2003).

REFERENCES

Wilcock, P.R., Crowe, J.C., 2003. Surface-based transport model for mixed-size sediment. J Hydraul Eng 129, 120–128. doi:10.1061/(ASCE)0733-9429(2003)129:2(120)

---

## Author Comment (AC1) · 8 Feb 2016

We thank the reviewer for his/her review. We fully agree with the reviewer in that it would have been better to conduct a larger series of experiments to assess the effect of the imposed flow and sediment conditions on the outcome of the laboratory experiment. There is an endless range of flow and sediment conditions under which the presented laboratory experiment could (and can) be repeated. In this study we had the opportunity to do one experiment, yet with a level of detail of the measurements that has not been shown before: we show with large detail the spatial and temporal changes

of the bed surface texture for a case of armor formation, breakup, and reformation. In particular, the novelties of our study are (a) a detailed data set on changes of the bed surface texture for a case of armor formation, breakup, and reformation under controlled conditions, (b) insights on the time scale of armor formation, (c) insights on the temporary increase of the sediment transport rate under conditions of armor breakup, (d) insights on the fact that a new armor formed quite rapidly rather than the formation of a local deep erosion pit, and (e) insights on the fact that the new armor was coarser than the initial one. Naturally the result of the laboratory experiment will be different under different flow conditions (e.g., Hassan et al., 2006, Guney et al., 2013) and different sediment conditions (e.g., Marion and Fraccarollo, 1997, Curran and Waters, 2014). Yet because of the above novelties the authors are convinced that the current results are worth sharing with the wider research community. Besides we expect that the current results will be helpful to researchers in setting up a more extensive series of laboratory experiments on armor studies (such as the series of experiments proposed by the referee), in illustrating how the image analysis technique is useful under such circumstances, and in avoiding practical issues associated with such experiments.

References

Curran, J., and K. A. Waters (2014), The importance of bed sediment sand content for the structure of a static armor layer in a gravel bed river, Journal of Geophysical Research: Earth Surface, 119 (7), 1484-1497, doi:10.1002/2014JF003143.

Guney, M., G. Bombar, and A. Aksoy (2013), Experimental study of the coarse surface development effect on the bimodal bed-load transport under unsteady ow conditions, Journal of Hydraulic Engineering, 139 (1), 12-21, doi:10.1061/(ASCE)HY.1943-7900.0000640.

Hassan, M. A., R. Egozi, and G. Parker (2006), Experiments on the effect of hydrograph characteristics on vertical grain sorting in gravel bed rivers, Water Resources

Research, 42, W09,408, doi:10.1029/2005WR004707.

Marion, A., and L. Fraccarollo (1997), Experimental investigation of mobile armoring development, Water Resources Research, 33 (6), 1447-1453, doi:10.1029/97WR00705.

**ESurfD**

---

## Referee Comment (RC2) · Anonymous Referee #2 · 10 Feb 2016

Review of Âń Armor breakup and reformation in a degradational laboratory experiment: detailed measurements of spatial and temporal changes of the bed surface texture" by Clara Orrú, Astrid Blom, and Wim S. J. Uijttewaal

GENERAL COMMENT

This paper presents the results of an armor breakup experiment, under condition of low sediment supply and changing hydraulics. The topic is of interest because armor is present and controls the morphodynamics in many rivers, and because the physics of armoring is complex and still largely misunderstood. Flume experiment is a good

way to approach the processes involved.

Before reviewing the paper, I had a look to the online discussion. I agree with referee 1 that the paper suffers from a series of drawbacks and can largely be improved. However I will not be as severe as he was, and I consider that it is a nice experiment which results deserve to be presented to the community, after major revision of the paper. Major comments My main comments concern the experimental set-up. When reading the paper the first time, I wondered if I missed something when I discovered the experimental set-up: why do you need a gravel sand transition for studying the armor breakup and reformation? This is an enigma. You could have done the same experiment without the sandy part of the flume? Or maybe this was motivated by a particular reason, but it is not explained in the paper. Actually your experiment is far to be out of interest, but what you studied seems more to be a gravel sand transition. This is a situation present in many rivers, which physics is also poorly understood. With your experiment you could describe precisely how the coexistence of armor and sand patches behave during flood: starvation and erosion of the sandy place, replacement by the gravel wave, impact of reduced slope in the propagation of the gravel wave and armor reformation. . .. Such experiment allows very well documented measurements, and we would expect a very fine analysis of the hydraulics. What were the hydraulics conditions: flow depth, velocity, energy slope, Fr, Shear stress, Shields stress. . .? Was there any side wall effects, and if yes can you propose a correction? The results you present in the paper are essentially descriptive. With the hydraulics, you could propose a much convincing quantitative analysis of your results, with a focus on what happens in the sand gravel transition zone. It would be very interesting. To conclude on this comment: either you justify the need of a sandy section for studying the armor breakup and reformation, of you reconsider the paper objective (which also means to reconsider the literature review). I also found a bit frustrating the description of the grain size measurement technique1.

SPECIFIC COMMENTS
P3 L3: how did you choose the flow conditions? P3 L15: I don't really understand what you mean by patches. Did you consider different grain size in each patch or did you use a trimodal mixture everywhere? P3 L16: I suppose that the energy slope was very different than the bed slope? P3 L19: could explain with a few lines? P3 L25-38: this part is very frustrating. A very nice equipment is presented in Fig4 but you don't really explain what it is. The method is not explained. What are these polygons? P4 L8-16: this aspect is particularly interesting. I don't know many papers describing in detail this situation. P4 L20-25 The armor breakup seems to concern the center of the flume? An evaluation of side wall effects would be interesting here. P5 L5-15: you should use the hydraulics (shear stress) to analyze these changes. Did you observe any regressive erosion at the gravel sand transition? P5 L16-28 This is an interesting result which deserves more comments

---

## Referee Comment (RC3) · Anonymous Referee #3 · 19 Feb 2016

General Comments

This paper describes a laboratory experiment where armor was built on an initial condition, then broken up and reformed. The paper describes an interesting experiment, but felt incomplete in that the methods are included elsewhere in a submitted paper (not accessible as far as I know), the discussion and conclusion were very brief and does not quite relate the results to an implication in the real world, which I would hope for. The paper would benefit in clarity with some reorganization - it took me multiple reads to understand certain sections. Some sections I still do not understand. I offer some

suggestions for making the paper easier to understand. I recommend a major revision and addition of more detail and information before publication of this paper.

Specific comments

Abstract I would like to see mentioned in the abstract something about the initial bed condition. It is spatially varying and that is important information.

I don't know exactly what is meant by "closer to normal flow conditions"

Section 2 experimental setup It would make much more sense to me to move paragraph 2, beginning "An initial experiment" to the end of the section after describing the sediment mixture and initial bed.

It would be useful to mention in the text that experiment T1 goes from -16 to 0 hours and T2 goes from 0 to 4.

In Figure1 it is not clear where "downstream" is in "downstream water surface elevation." Do you mean "sand reach"?

Section 2.2 measurements

I'm having a hard time saying it is okay to review and accept this paper before seeing and evaluating (Orru, submitted 2015) is available to view and evaluate. I know this is a difficult situation if that review is taking longer than expected, but I cannot evaluate the methodology that is the basis for all of the results. The description here should be more complete, instead of assuming one can read Orru, submitted 2015.

Section 2.3 Because the bed is spatially varying, it is important to say where things happened. Where did the armor form? Say it in the text, and label the armor section in Figure 5, even if you think it is obvious.

Last sentence of 2.3 "Armor was considered fully developed after 16 h" (note that this is "0 hr" in this paper's figures, etc.)

The described bed step is curious to me. Does this have any relevance to nature or "mess up" any of the interpretation of the lab results.

Section 3 "increasing the water discharge" - please state increased from what to what, even though it is in the figure, it is good to put it in the text. ("increase by 25%" or something like that) Give the reader an idea of how much it was increased.

Section 3.1's first paragraph very hard to follow. "Yet the fining was even stronger than we measured" - Why? Suggest rephrasing this paragraph and asking around if it makes sense to colleagues.

Figure 8 - the points vary in streamwise coordinate in the figure (there is spread in the x-axis)- are they supposed to represent one point in the streamwise coordinate? I was a bit confused by this.

Section 3.2 It is not clear to me how does figure 10 show lateral variation in degradation?

Not much is reported about the sediment transport captured in the sediment trap?

Section 4 discussion

Some of these sentences could be rearranged for better understanding.

I don't know what is the field case in "in the field case"?

In general the discussion seems short and confusing, and could be improved by providing implications for natural streams. Yes, comparisons to other studies were given, but not really related back to nature. By adding more content and having a better narrative, the discussion could be really improved.

The conclusions section is similarly hard to follow. There should make some mention of the initial bed condition and what was "base flow". Is there an implication to the last sentence?
**ESurfD**

Interactive
comment

---

## Author Comment (AC2) · 2 Mar 2016

The authors would like to thank Anonymous Referee #2 for his/her clear and constructive comments.

The experimental set-up was chosen following an earlier laboratory experiment on armor formation over a bed with an initial streamwise fining pattern. The main objective of that study was to test a new image analysis technique for detailed measurements of the bed surface texture during flow in a sand-gravel laboratory experiment. The study considers a laboratory experiment that is different from (but similar to) the one we describe in the current manuscript as in that study the sediment mixture was bimodal and the gravel fraction was much coarser and fully immobile (partial mobility). That study is described in a manuscript that is currently under review with Water Resources Research. We observed how a reach that was characterized by an initially gradual fining pattern under a limited sediment supply rate developed into a more abrupt spatial transition in grain size and slope. Partial transport conditions prevented the adjustment of the bed and the approach to normal flow conditions (i.e., a backwater was present in the equilibrium state).

We are aware that the results of the laboratory experiment suggest a similarity with a gravel-sand transition. Yet, we are hesitant to claim a similarity between the mechanisms in our experiment and a gravel-sand transition in the field since in the experiment we excluded many aspects that have been recognized to be relevant to the development of a gravel-sand transition, such as the progradation of a gravel wedge (Paola et al., 1992; Seal et al., 1997), basin subsidence (Paola, 1988; Parker and Cui, 1998), base level change (Pickup and Warner, 1984; Sambrook Smith and Ferguson, 1995), abrasion (Yatsu, 1955; Parker and Cui, 1998), and suspended load transport (Venditti and Church, 2014; Venditti et al., 2015). We have included this argumentation in the WRR manuscript and we will include it shortly in the revised version of the current manuscript.

The current manuscript shows similar results to the WRR manuscript with respect to only the initial phase of our experiment, i.e., the formation of the armor, which is described in Section 2.3 of the current manuscript. Yet in the current study our focus is on studying the mechanisms of armor breakup and its consequences, which have not been treated in the WRR manuscript. We apologize for the fact that we have provided insufficient information on the image analysis technique, as well as on the hydraulic conditions. In the above WRR manuscript we describe these points in close detail, and we did not want to repeat that part in this paper. The WRR manuscript specifically aims at describing the image analysis technique and its results and so addresses the

**ESurfD**
issues raised by the current reviewer related to the image analysis technique. The WRR manuscript also focusses on the hydraulic conditions of a different (yet similar) laboratory experiment, and on reproducing that laboratory experiment with a numerical model. We did not realize that the limited information in the current manuscript on the image analysis technique and the hydraulic conditions have made the current manuscript on armor breakup insufficiently stand-alone. We propose that we revise the current manuscript such that it becomes stand-alone and we will provide the reviewers with access to the WRR manuscript.

References

Paola, C. (1988), Subsidence and gravel transport in alluvial basins, in New perspectives in basin analysis, pp. 231-243, Springer.

Paola, C., G. Parker, R. Seal, S. K. Sinha, J. B. Southard, and P. R. Wilcock (1992), Downstream fining by selective deposition in a laboratory flume, Science, 258 (5089), 597 1757-1760, doi:10.1126/science.258.5089.1757.

Parker, G., and Y. Cui (1998), The arrested gravel front: stable gravel-sand transitions in rivers part 1: Simplified analytical solution, Journal of Hydraulic Research, 36 (1), 75-100, doi:10.1080/00221689809498379.

Pickup, G., and R. Warner (1984), Geomorphology of tropical rivers II. Channel adjustment to sediment load and discharge in the Fly and lower Purari, Papua New Guinea, Catena Supplement, 5, 18-41.

Sambrook Smith, G. H., and R. I. Ferguson (1995), The gravel-sand transition along river channels, Journal of Sedimentary Research, 65 (2).

Seal, R., C. Paola, G. Parker, J. Southard, and P. R. Wilcock (1997), Experiments on downstream fining of gravel I: Narrow-channel runs, Journal of Hydraulic Engineering, 123 (10), 874-884, doi:10.1061/(ASCE)0733-9429(1997)123:10(874).

Venditti, J. G., and M. Church (2014), Morphology and controls on the position of

a gravel-sand transition: Fraser river, British Columbia, Journal of Geophysical Research: Earth Surface, 119 (9), 1959-1976, doi:10.1002/2014JF003147.

Venditti, J. G., N. Domarad, M. Church, and C. D. Rennie (2015), The gravel-sand transition: Sediment dynamics in a diffuse extension, Journal of Geophysical Research: Earth Surface, 120 (6), 943-963, doi:10.1002/2014JF003328.

Yatsu, E. (1955), On the longitudinal profile of the graded river, Eos, Transactions American Geophysical Union, 36 (4), 655-663, doi:10.1029/TR036i004p00655.

**ESurfD**

---

## Author Comment (AC3) · 2 Mar 2016

The authors thank Referee #3 for his/her helpful and constructive review.

We apologize for the fact that we have provided insufficient information on the image analysis technique. A manuscript presenting the new measurement technique for measuring the bed surface texture in detail is currently under review with Water Resources Research. We did not want to repeat the information but realized insufficiently that the current manuscript is therefore not sufficiently stand-alone. We propose that we revise the current manuscript such that it does become stand-alone and we will pro-

vide the referees with access to the submitted WRR manuscript. We thank Referee #3 for his/her detailed suggestions on improvement of the structure of the manuscript and his/her requests for further explanation, which will help improve the clarity of the manuscript.

With normal flow conditions we indicate a state in which the slope of the water surface is equal to the slope of the channel bed. If the boundary conditions of a reach (i.e., upstream water discharge and sediment supply rate, as well as the downstream water surface elevation) are constant for a sufficiently long time, the reach will approach normal flow conditions (provided that particle abrasion and tributaries do not play a role). Yet, conditions of partial transport, in which the coarse fractions of the sediment are immobile, and the associated armor can prevent this adaptation of the bed and its approach to normal flow conditions. Armor breakup then enables adjustment of the bed slope such that the bed slope is closer to the water surface slope and the final bed configuration is closer to normal flow. The sediment contributing to the migration of the front of the gravel reach appeared to provide much more information regarding the local increase of the sediment transport capacity due to armor breakup than the sediment caught in the sand trap. However it is a good idea if in Figure 11 we not only report the sediment transport capacity at the gravel reach front but also the sediment transport capacity at the downstream end of the flume.

The bed step has formed as a result of the adjustment of the bed to the limited sediment supply. As no sediment is fed at the upstream end of the flume, the bed approaches a final state that is characterized by zero sediment transport. This situation of zero transport results from either (a) the washing out of fines from the bed surface or (b) if fines are present at the bed surface they must be characterized by a Shields stress that is smaller than the critical one. For the sand reach, a state of zero sediment transport is governed by a much smaller flow velocity (and so larger flow depth) than for the upstream gravel reach, which results in the observed bed step.

**ESurfD**

Interactive
comment

---

## Author Response (AR1)

**Response to the Reviewers' comments**

**Armor breakup and reformation in a degradational laboratory experiment**

C. Orrú, A. Blom, W.S.J. Uijttewaal

**Anonymous Referee #1**

*SUMMARY*

*This paper examines how the texture of a mixed sand-gravel channel bed responds to changes in flow discharge. It is qualitatively interesting to see the formation of static armor during high flow followed by breakup of this armor and reformation of a coarser mobile armor during subsequent high flow. Furthermore, application of new techniques for repeated longitudinal profiles of grain size and bed elevation offers new quantitative insight into armoring processes. While some of the observations are interesting, I think they are insufficient to merit publication. Though it was very long (20 h), only one experiment was run. Therefore, it is impossible to determine the reproducibility of these results or the dependence of the armor formation and breakup processes on the specific initial bed configuration, pattern of flow changes, and trimodal bed texture. Additional experiments to vary at least one of the experimental variables (e.g., magnitude of low and high flow) would provide much more insight into the controlling factors of bed armoring. Furthermore, the manuscript is poorly organized and needs to be fundamentally restructured. The abstract and conclusion are nearly identical and do not establish the motivation and implications for this work. The introduction is very disorganized and does not follow a logical progression in presenting information about past work. For example, the opening sentence of the paragraph at line 18 on page 1 seems unrelated to the remaining content in the paragraph. Some information in the methods (section 2) really belongs in the results (section 3); e.g., section 2.3. In the results, there are several assertions made without proper explanation. For example, on page 4, line 14- 15, an "imbricated structure" is mentioned, but there is never any discussion of when the imbrication developed or what morphological features suggest this interpretation. Related to this, there are several interpretations in the results (section 3), which really belong in the discussion (section 4); e.g., section 3.2, lines 9-10. Finally, all figures should be mentioned in the manuscript (currently Fig. 2 is missing), and the figure numbering should correspond to the order of mention (currently Fig. 10 is mentioned before Fig. 6).*

*Given these concerns, I therefore recommend rejection for this manuscript. I encourage the authors to run additional experiments, then more clearly present their results for future publication of this work. Further minor comments are listed below.*

We thank Reviewer #1 for his/her comments. We have revised the Abstract, Introduction, Discussion and Conclusion sections. Section 2.3 has been moved to the Results section. We have added more information on the imbricated structure. Section 3.2 (lines 9-10) has been moved to the Discussion. Figure 1 has now been mentioned.

*MINOR COMMENTS*

*1) In the abstract, the phrasing, "trimodal mixture composed of sand and gravel," implies that there are three components, but only two are mentioned here. It would be better just to list the three sizes (1, 6, and 10 mm) here.*

We have specified the mixture and the grain sizes in the abstract (Page 1, Lines 1-2).

*2) Page 3, line 13-17. Why do you impose a stepwise fining pattern in the initial bed? How does this reveal the dynamics of static and mobile armor formation better than a uniformly graded bed? Please explain in the manuscript. Also, the description of patch lengths is confusing. You should refer here to Fig. 2, which is never mentioned in the text.*

We have clarified the choice of the initial stepwise fining pattern in the Introduction (Page 2-3, Lines 35, 1-3). We have provided more information on the compartments more precisely and now refer to Figure 1 (Page 3, Lines 25-29).

*3) Section 2.2 on grain size techniques is insufficient to understand these techniques. Unless you have a strong reason to delve into the details, I suggest summarizing this to one or two lines then referring to Orru et al (submitted 2015) for further information. I also suggest removing Figures 3 and 4 for this reason.*

We have revised this part of the text and added information on the image analysis technique and the equipment applied to take images of the bed surface (Page 4, Lines 15-28).

*4) In describing the armor breakup and reformation, it is very difficult to see these bed changes from image to image in Figure 7. Could you apply some kind of image differencing technique to make the changes more apparent? The quantitative information in Figs. 8 and 9 is much more useful. Finally, unless you can justify them with more quantitative information, I would suggest removing the assertions on page 4, line 27-29, about textural changes between grain size analyses.*

The images in Figure 8 are snapshots from a video. The sand in transport in the parts indicated by the arrows makes part of the images (a) and (b) a bit blurry. The fact that image (c) indicates that there was no more transport of sand and the armor has reformed.

*5) The methods used for characterizing the bedload transport rate from the front propagation are confusing (page 5 and Fig. 11). Where is the front being measured, in terms of the streamwise coordinate? Is there really no observed transport here prior to the step increase in discharge (as indicated in Fig. 11), or are you just assuming this?*

The front progadation is measured at the crest. Before the increase in discharge, sediment was transported over the front (Figure 12 b) but this did not lead to front progradation.

*6) At page 5, line 26, it is not clear whether the mentioned bed load transport rate is referring to the propagating front bed load transport or some other measurement. It is curious that a bed load trap is mentioned on page 2, lines 32-33, but then never again mentioned in the paper.*

Thank you. We have now included information about the sediment transport measured at the downstream end of the flume (Figure 12, Page 6-7, Lines 32-34, 1-13). We have analysed these results in more detail.

*7) Finally, I am a bit concerned that no mention of the shear stress or Shields parameter is made here, despite the fact that this is usually considered an important variable in studies of the evolution of mixed sediment surfaces and armoring (e.g., Wilcock and Crowe, 2003).*

Thank you. We have now added an analysis of the Shields stress (Figure 11, Page 6 Lines 23-31).

**Anonymous Referee #2**
**GENERAL COMMENT**
*This paper presents the results of an armor breakup experiment, under condition of low sediment supply and changing hydraulics. The topic is of interest because armor is present and controls the morphodynamics in many rivers, and because the physics of armoring is complex and still largely misunderstood. Flume experiment is a good way to approach the processes involved. Before reviewing the paper, I had a look to the online discussion. I agree with referee 1 that the paper suffers from a series of drawbacks and can largely be improved. However I will not be as severe as he was, and I consider that it is a nice experiment which results deserve to be presented to the community, after major revision of the paper.*
*Major comments*
*My main comments concern the experimental set-up. When reading the paper the first time, I wondered if I missed something when I discovered the experimental set-up: why do you need a gravel sand transition for studying the armor breakup and reformation? This is an enigma. You could have done the same experiment without the sandy part of the flume? Or maybe this was motivated by a particular reason, but it is not explained in the paper. Actually your experiment is far to be out of interest, but what you studied seems more to be a gravel sand transition. This is a situation present in many rivers, which physics is also poorly understood. With your experiment you could describe precisely how the coexistence of armor and sand patches behave during flood: starvation and erosion of the sandy place, replacement by the gravel wave, impact of reduced slope in the propagation of the gravel wave and armor reformation. Such experiment allows very well documented measurements, and we would expect a very fine analysis of the hydraulics. What were the hydraulics conditions: flow depth, velocity, energy slope, Fr, Shear stress, Shields stress? Was there any side wall effects, and if yes can you propose a correction? The results you present in the paper are essentially descriptive. With the hydraulics, you could propose a much convincing quantitative analysis of your results, with a focus on what happens in the sand gravel transition zone. It would be very interesting. To conclude on this comment: either you justify the need of a sandy section for studying the armor breakup and reformation, of you reconsider the paper objective (which also means to reconsider the literature review). I also found a bit frustrating the description of the grain size measurement technique.*

We thank Reviewer #2 for his/her suggestions. We have clarified the choice of the initial stepwise fining pattern in the Introduction and we have included the argumentation about the similarities with a gravel-sand transition in the current manuscript (Page 3, Lines 4-11).

**SPECIFIC COMMENTS**

*8) P3 L3: how did you choose the flow conditions?*
Flow conditions were chosen such that we obtained the desired sediment mobility, partial transport conditions in order to create the armor in Experiment T1, which was based also on the previous experiment presented in

the WRR manuscript, and fully mobile conditions in Experiment T2. In addition, a certain minimum flow depth is required to not significantly affect the flow when using the floating device (Figure 4) to measure the grain size distribution of the bed surface during the experiment.

*9) P3 L15: I don't really understand what you mean by patches. Did you consider different grain size in each patch or did you use a trimodal mixture everywhere?*

We meant compartments of the initial bed characterized by a different grain size distribution. The compartments were filled with different volumetric fraction contents of the 3 grain size fractions. The sand content increased in streamwise direction in steps of 10 percent for each compartment. We have specified the characteristics of the compartments more precisely (Page 3, Lines 28-29).

*10) P3 L16: I suppose that the energy slope was very different than the bed slope?*

Figure 1 shows a comparison between the bed slope and the energy slope during Experiments T1 and T2. The large peak in the bed slope indicates the front of the trimodal reach. One can distinguish the migration of the front. In the initial part of Experiment T1 the bed and energy slope were comparable and later they started to differ due to the bed degradation mainly over the sand reach and the presence of the backwater curve. During Experiment T2 the bed and energy slope were comparable only in the upstream trimodal reach where conditions were closer to normal flow.

[Figure]

**Figure 1 Comparison of bed slope and energy slope during the Experiments T1 and T2.**

*11) P3 L19: could explain with a few lines? P3 L25-38: this part is very frustrating. A very nice equipment is presented in Fig4 but you don't really explain what it is. The method is not explained. What are these polygons?*

We have added some explanation on the measurements of water discharge and bed and water surface elevations (Page 4, Lines 7-11). We have added information on the image analysis technique and the equipment used to take images of the bed surface (Page 4, Lines 15-28). We removed the information on the polygons and now refer to the WRR manuscript.

*12) P4 L8-16: this aspect is particularly interesting. I don't know many papers describing in detail this situation.*

Thank you. We have treated this aspect in more detail in the WRR manuscript, and we have more clearly stressed it in the current manuscript.

*13) P4 L20-25 The armor breakup seems to concern the center of the flume? An evaluation of side wall effects would be interesting here.*

The armor breakup seemed to occur randomly in streamwise and lateral direction. It did not concern only the center of the flume. As mentioned in the Discussion section one of the hypotheses is that besides the increase in flow rate, irregularities, randomly distributed over the bed surface, seemed to initiate the breakup due to turbulence.

*14) P5 L5-15: you should use the hydraulics (shear stress) to analyze these changes. Did you observe any regressive erosion at the gravel sand transition?*

We now present an analysis of the hydraulic conditions (Figure 11, Page 6, Lines 23-31). We did not observe regressive erosion during the experiment.

*15) P5 L16-28 This is an interesting result which deserves more comments.*

We revised this part and we have added more information (Page 6-7, Lines 32-34, 1-13).

**Anonymous Referee #3**
*GENERAL COMMENT*
*This paper describes a laboratory experiment where armor was built on an initial condition, then broken up and reformed. The paper describes an interesting experiment, but felt incomplete in that the methods are included elsewhere in a submitted paper (not accessible as far as I know), the discussion and conclusion were very brief and does not quite relate the results to an implication in the real world, which I would hope for. The paper would benefit in clarity with some reorganization - it took me multiple reads to understand certain sections. Some sections I still do not understand. I offer some suggestions for making the paper easier to understand. I recommend a major revision and addition of more detail and information before publication of this paper.*

*SPECIFIC COMMENTS*

*16) Abstract I would like to see mentioned in the abstract something about the initial bed condition. It is spatially varying and that is important information.*

We now mention the initial bed condition in the abstract (Page 1, Lines 2-4).

*17) I don't know exactly what is meant by "closer to normal flow conditions".*

We have added information about normal flow conditions in the text (Page 6, Lines 15-22).

*18) Section 2 experimental setup It would make much more sense to me to move paragraph 2, beginning "An initial experiment" to the end of the section after describing the sediment mixture and initial bed.*

We have moved this paragraph to the end of the section (Page 3- 4, Lines 31-33, 1-5).

*19) It would be useful to mention in the text that experiment T1 goes from -16 to 0 hours and T2 goes from 0 to 4.*

We have now specified the time intervals in the text (Page 4, Lines 2-5).

*20) In Figure1 it is not clear where "downstream" is in "downstream water surface elevation." Do you mean "sand reach"?*

The water surface elevation was measured at the downstream end of the flume (x = 10.62 m). We have added this information to the caption of Figure 2.

*21) Section 2.2 measurements*
*I'm having a hard time saying it is okay to review and accept this paper before seeing and evaluating (Orru, submitted 2015) is available to view and evaluate. I know this is a difficult situation if that review is taking longer than expected, but I cannot evaluate the methodology that is the basis for all of the results. The description here should be more complete, instead of assuming one can read Orru, submitted 2015.*

We have now clarified and added information on the image analysis technique and the equipment used to take pictures of the bed surface (Page 4, Lines 15-28). We provide the pdf of the manuscript that is currently under review with WRR to the reviewers.

*22) Section 2.3 Because the bed is spatially varying, it is important to say where things happened. Where did the armor form? Say it in the text, and label the armor section in Figure 5, even if you think it is obvious.*

We have mentioned where the armor formed in the text (Page 5, Lines 3) and now indicate the zone in Figure 5.

*23) Last sentence of 2.3 "Armor was considered fully developed after 16 h" (note that this is "0 hr" in this paper's figures, etc.).*

Thank you, we have added this information (Page 5, Lines 17).

*24) The described bed step is curious to me. Does this have any relevance to nature or "mess up" any of the interpretation of the lab results.*

We have clarified the formation on the step in bed elevation in the text (Page 5, Lines 5-14).

*25) Section 3 "increasing the water discharge" - please state increased from what to what, even though it is in the figure, it is good to put it in the text. ("increase by 25%" or something like that) Give the reader an idea of how much it was increased.*

We have added this information (Page 5, Lines 21-22).

*26) Section 3.1's first paragraph very hard to follow. "Yet the fining was even stronger than we measured" - Why? Suggest rephrasing this paragraph and asking around if it makes sense to colleagues.*

We have clarified this paragraph (Page 5, Lines 25-33).

*27) Figure 8 - the points vary in streamwise coordinate in the figure (there is spread in the x-axis)- are they supposed to represent one point in the streamwise coordinate? I was a bit confused by this.*

Thank you, we revised Figure 9.

*28) Section 3.2 It is not clear to me how does figure 10 show lateral variation in degradation?*

Figure 6 and our measurements do not show lateral variation in degradation, yet this was only observed by the author. We have now avoided referring to Figure 6 to avoid confusion.

*29) Not much is reported about the sediment transport captured in the sediment trap?*

We have now included information about the sediment transport measured at the downstream end of the flume (Figure 12, Page 6-7, Lines 32-33, 1-13).

*30) Section 4 discussion*
*Some of these sentences could be rearranged for better understanding. I don't know what is the field case in "in the field case"?*

We have specified the reference (Page 8, Lines 13-14).

*31) In general the discussion seems short and confusing, and could be improved by providing implications for natural streams. Yes, comparisons to other studies were given, but not really related back to nature. By adding more content and having a better narrative, the discussion could be really improved.*

We have revised the Discussion section and we have discussed implications of the current study to rivers (Page 7, Lines 15-22).

*32) The conclusions section is similarly hard to follow. There should make some mention of the initial bed condition and what was "base flow". Is there an implication to the last sentence?*

We have revised the Conclusions section mentioning the initial bed conditions and the implication of 
[revised manuscript text omitted]